# Effect of Enhanced Recovery After Surgery (ERAS) Implementation on Postoperative Atrial Fibrillation in Cardiac Surgery

**DOI:** 10.3390/biomedicines13051212

**Published:** 2025-05-16

**Authors:** Romain Niessen, Valentina Rancati, Mario Verdugo-Marchese, Ziyad Gunga, Anna Nowacka, Valentine Melly, Christophe Abellan, Karima Alouazen, Tamila Abdurashidova, Caroline Botteau, Matthias Kirsch, Zied Ltaief

**Affiliations:** 1Department of Intensive Care Medicine, Lausanne University Hospital (CHUV), 1011 Lausanne, Switzerland; 2Department of Anesthesia, Lausanne University Hospital (CHUV), 1011 Lausanne, Switzerland; 3Department of Cardiac Surgery, Lausanne University Hospital (CHUV), 1011 Lausanne, Switzerland; 4Department of Cardio-Respiratory Physiotherapy, Lausanne University Hospital (CHUV), 1011 Lausanne, Switzerland; 5Faculty of Biology and Medicine, University of Lausanne (UNIL), 1011 Lausanne, Switzerland

**Keywords:** atrial fibrillation, postoperative atrial fibrillation, enhanced recovery after surgery, cardiac surgery, perioperative medicine

## Abstract

**Background/Objectives**: Postoperative atrial fibrillation (POAF) is the most frequent arrhythmic complication following cardiac surgery and is associated with increased morbidity and prolonged recovery. This study aimed to evaluate the impact of an enhanced recovery after surgery (ERAS) program on the incidence of POAF and broader perioperative outcomes. **Methods**: In this monocentric, observational cohort study, we compared a retrospective pre-ERAS cohort (*n* = 162) with a prospective ERAS cohort (*n* = 321). The primary outcome was the incidence of POAF, assessed using two definitions: (1) the American Association for Thoracic Surgery (AATS) 2014 clinical definition, identifying POAF as atrial fibrillation requiring treatment; and (2) the European Society of Cardiology (ESC) 2024 definition, describing new-onset atrial fibrillation occurring immediately after surgery. Secondary outcomes included compliance with POAF prophylaxis measures, length of hospital stay, and the occurrence of postoperative complications. Statistical analyses included propensity score matching and multivariate logistic regression to identify independent predictors of POAF. **Results**: ERAS implementation was associated with a significant reduction in POAF incidence across both definitions. According to the AATS 2014 definition, POAF occurred in 20% of ERAS patients vs. 39% in the pre-ERAS group (*p* = 0.001), and 23% vs. 39% in the matched cohort (*p* = 0.004). Using the ESC 2024 definition, POAF was observed in 21% vs. 37% (*p* = 0.001) in unmatched and 20% vs. 36% (*p* = 0.005) in matched populations. Compliance with POAF prophylaxis improved markedly in the ERAS group (70% vs. 21%, *p* = 0.001). ERAS patients also experienced shorter hospital stays and fewer postoperative complications (26% vs. 38% in the matched cohort, *p* = 0.033). **Conclusions**: The implementation of a structured ERAS protocol significantly reduced POAF incidence, improved compliance with preventive strategies, and enhanced key aspects of postoperative recovery.

## 1. Introduction

New-onset atrial fibrillation (AF) after surgery, commonly called postoperative atrial fibrillation (POAF), is the most common rhythmic complication after cardiac surgery and also the most common type of secondary AF [1,2]. Its incidence ranges from 30% to 50%, depending on cardiac surgery type [1,2]. The definitions used vary, ranging from “any new-onset POAF” [3] to “any POAF requiring treatment” [4], “any POAF episode lasting more than 30 s” [5], or “any POAF lasting longer than 10 min” [6]. Persistent POAF can lead to an increase in morbidity, mortality, intensive care unit (ICU) stay duration, hospital stay duration, and hospital costs [1,7,8]. Moreover, POAF is associated with a five-fold increase in recurrent AF in the next five years, with increased risks of stroke, myocardial infarction, heart failure, and mortality [9,10,11]. The association between POAF and long-term death after cardiac surgery has been highlighted in a recent meta-analysis with 10 studies (44,367 patients), showing that patients with POAF had more than twice the risk of 1-year mortality compared to those without [12]. Patient-related risk factors for POAF are advanced age, male gender, white non-Hispanic race, arterial hypertension, myocardial infarction history, heart failure, previous AF, unrecognized severe obstructive sleep apnea syndrome, chronic obstructive pulmonary disease, smoking, dyslipidemia, obesity, diabetes, impaired renal function, and dysnatremia [13,14,15]. Surgical insult favors POAF occurrence due to direct atrial tissue injury, such as right atriotomy for venous cannulation or mitral and tricuspid surgery [8,16]. In the postoperative period, POAF triggers include systemic inflammatory reactions due to cardiopulmonary bypass (CPB), surgical trauma and retained blood syndrome [17], dyselectrolytemia, such as hypokalemia or hypomagnesemia, and intra-cavitary catheters [16]. Moreover, sympathetic nervous system hyperactivity, high pain-induced catecholamine levels, delirium, and inotropic agents increase the risk of POAF [18].

Multiple strategies to prevent POAF have been tested based on AF risk factors and underlying pathophysiology, but evidence from randomized controlled trials (RCTs) or pilot studies remains limited [19,20]. Currently, guidelines recommend the systematic use of beta-blockers and amiodarone during the perioperative period [21,22]. Non-pharmacological interventions such as epicardial pacing, a posterior pericardiotomy, and posterior pericardial drainage have shown interesting results for reducing POAF incidence [5,23,24,25,26].

The cardiac surgery program at Lausanne University Hospital (CHUV) received ERAS^®^ Society certification in October 2023. As part of this certification process, we revised our protocols for preventing POAF to implement a comprehensive, evidence-based approach. This study aims to evaluate the effectiveness of ERAS protocols in reducing the incidence of POAF following cardiac surgery.

## 2. Materials and Methods

This monocentric study, conducted in accordance with the Declaration of Helsinki and adhering to the STROBE guidelines, was approved by the Ethics Committee of the Commission cantonale d’éthique de la recherche sur l’être humain du Canton de Vaud (CER-VD #2024-00632). It compares a retrospective cohort of patients who underwent elective cardiac surgery at Lausanne University Hospital (CHUV) in 2019, before the implementation of ERAS protocols, with a prospective cohort of patients benefiting from certified cardiac ERAS protocols starting in 2023. Data for the ERAS cohort were collected prospectively following certification, with variables predefined before ERAS implementation. Information was gathered from the hospital’s electronic medical records and entered into the Research Electronic Data Capture (REDCap) database.

### 2.1. Participants

The cardiac ERAS program was initiated at Lausanne University Hospital in May 2023, with patient enrollment continuing until February 2025. To provide a reliable comparison, we selected patients from 2019 who met the inclusion criteria, as it was the last year unaffected by the COVID-19 pandemic. Eligible patients were adults over 18 years old who underwent elective cardiac surgery via median sternotomy with CPB, regardless of surgical complexity or history of prior cardiac surgery. Exclusion criteria included patients undergoing urgent surgeries, ventricular assist device implantation, heart transplantation, or those who did not consent to the use of their data.

### 2.2. ERAS Protocols

Our institution benefits from a certification process, which has previously been published [27]. The ERAS^®^ Society requires well-defined, recognized protocols and adherence rates > 70% to receive the certification. These protocols rely on the ERAS^®^ Cardiac Society guidelines published by Engelman et al. [28] and additional recommendations, as well as the department’s pre-existing perioperative management practices. At the CHUV, the ERAS program for cardiac surgery began in May 2023 and board certification from the ERAS Society was obtained in October 2023. The ERAS perioperative pathway consists of seven steps: surgical consultation, anesthesia and ERAS nurse consultations, preoperative hospitalization, surgery, ICU stay, intermediate care, and postoperative hospitalization. It aims to achieve eight key objectives: patient education and empowerment, optimization of nutrition and glycemic control, effective anemia management through the patient blood management program, infection prevention, minimizing invasive ventilation and sedation, implementing goal-directed hemodynamic therapy, reducing pain with multimodal analgesia and minimizing opioid use, and promoting early mobilization with the timely removal of drains and equipment. This comprehensive approach also includes measures such as hypothermia prevention, delirium detection and prevention, preoperative POAF prevention, and early nutrition to enhance recovery and reduce postoperative complications. All efforts are directed towards reducing operative time, CPB duration, aortic clamping time, mechanical ventilation duration, and length of stay in both the ICU and the hospital [27,29].

All patients underwent cardiac surgery via median sternotomy; no minimally invasive or robotic-assisted approaches were used during the study period.

A multimodal systemic analgesia protocol, combined with local infiltration or a parasternal intercostal plane block at the surgical site, is used to relieve pain and decrease pain-induced elevations in catecholamine levels. This, in turn, impacts the following.

The patient blood management (PBM) program is an integral part of the ERAS protocol. As soon as open-heart surgery is scheduled, a preoperative anemia work-up, including ferritin, transferrin saturation, and creatinine, is performed. Deficiencies are treated with intravenous iron, vitamin B12, and folic acid. If renal anemia is diagnosed, the patient is treated with erythropoietin alpha (EPOα), vitamin B12, and folic acid. During surgery, a restrictive transfusion threshold of Hb ≥ 70 g/L is maintained using an autologous blood recovery system or heterologous blood transfusion. In the postoperative period, an Hb level ≥ 70 g/L is maintained, except if transfusion triggers such as chest pain, electrocardiographic signs of myocardial ischemia, elevated serum lactate, or central venous oxygen saturation (ScvO_2_) below 70% are identified.

The methodology of the POAF prevention protocol includes preoperative, intraoperative, and postoperative strategies. Preoperatively and in the absence of contraindications, patients without beta-blocker therapy are treated with metoprolol (25 mg once daily) on day 1 and on surgery day. Patients already on beta-blockers continue their usual regimen until surgery day. Intraoperatively, all patients receive 4 g of IV magnesium after anesthesia induction. Peroperatively, close monitoring of temperatures and active warming if needed prevents hypothermia. Posterior pericardial drainage is favored. Multiple blood analyses are performed, and potassium levels are closely monitored and corrected if needed. Postoperatively, beta-blockers are resumed as early as possible if they are not contraindicated, maintained until day 5, and stopped if no further indication exists. Serum magnesium and potassium levels are monitored daily and supplemented orally if needed.

### 2.3. Data Collection

Collected baseline characteristics included age, gender, body mass index (BMI), smoking status, alcohol consumption, recreational drug use, arterial hypertension, diabetes, dyslipidemia, chronic pulmonary disease, extracardiac arteriopathy, cerebrovascular disease, European System for Cardiac Operative Risk Evaluation (EuroSCORE) II, and left ventricular ejection fraction (LVEF). Additionally, surgical variables recorded comprised the type of surgery, specific procedures performed, aortic cross-clamp duration, and CPB duration. Furthermore, data were collected on the incidence of POAF. Prophylactic measures such as preoperative beta-blocker use and reintroduction after surgery were systematically documented to assess protocol efficacy. Data collection also included various postoperative outcomes and recovery metrics. These comprised the duration from surgery to discharge, the duration of the ICU stay, the extubation rate in the operating room, the extubation time in the ICU, opioid use from postoperative day (POD) 0 to POD 3 (expressed as morphine milligram equivalent (MME)), early mobilization for the first meal post-extubation, mobilization on POD 1, time to bowel function recovery, and the incidence of postoperative complications. Complications were defined by criteria including 30-day mortality, reoperation, ICU readmission, acute confusional state, postoperative stroke, myocardial infarction, hospital-acquired pneumonia (HAP), hospital-acquired infections, pleural effusion requiring intervention, and acute kidney injury (AKI) requiring dialysis. The proportion of patients free from these complications was also documented to evaluate overall protocol effectiveness.

### 2.4. Objectives and Outcomes

The primary outcome of this study is to evaluate the impact of the ERAS program on the incidence of POAF using two distinct definitions: (1) the AATS 2014 clinical definition, which identifies POAF as atrial fibrillation occurring in the intraoperative or postoperative setting that requires treatment with rate or rhythm control agents and anticoagulation, or which leads to an extended hospital stay, and (2) the ESC 2024 definition, describing POAF as new-onset atrial fibrillation occurring immediately after surgery. Secondary outcomes include assessing compliance with the ERAS protocol, particularly regarding prophylactic measures against POAF, and evaluating the broader impact of the ERAS program on the patient’s overall clinical pathway, including the length of hospital stay and the occurrence of postoperative complications.

### 2.5. Statistical Analysis

The statistical analysis was performed to evaluate the impact of the ERAS program on the occurrence of POAF. The two groups (ERAS vs. non-ERAS) were first compared using standard univariate statistical methods. Matching was then performed using propensity score matching (PSM) to balance baseline characteristics, including age and EuroSCORE II, between patients managed with the ERAS program and those without. Matching was achieved using a 1:1 nearest-neighbor matching algorithm without replacement to reduce potential confounding. The effect of the ERAS program on the new onset of POAF was assessed using a multivariate logistic regression model, adjusted for age, gender, type of surgery, EuroSCORE II, and coronary artery bypass grafting (CABG) as the main procedure. Results from the multivariate analysis were reported as odds ratios (ORs) with a 95% confidence interval (CI) and visualized in a forest plot. Continuous variables were expressed as medians and interquartile ranges (IQRs), while categorical data were presented using absolute numbers and percentages. Continuous variables were compared using the Student’s *t*-test for normally distributed data and the Mann–Whitney U test for non-normally distributed data. For categorical variables, associations were assessed using the chi-squared test. All statistical tests were two-tailed, with a significance threshold set at *p* < 0.05. Analyses were conducted using Stata Version 18.0 (StataCorp LLC, TX, USA).

## 3. Results

The flowchart of this study is presented in Figure 1 and depicts the selection process, beginning with all patients undergoing cardiac surgery in 2019 and the ERAS cohort. After applying inclusion and exclusion criteria, 321 patients were allocated to the ERAS group and 162 to the pre-ERAS group, totaling 483 individuals. Propensity score matching was then performed, resulting in a final matched cohort comprising 145 patients in the ERAS group and 145 patients in the pre-ERAS group for comparative analysis.

### 3.1. Characteristics of the Population

The demographic characteristics and surgery data of the full and matched populations are presented in Table 1. In the unmatched cohort, significant differences were observed between the ERAS and pre-ERAS groups concerning age and risk assessment. Patients in the ERAS group were younger, with a median age of 64.0 years [IQR: 56.0; 71.0] compared to 67.0 years [IQR: 59.0; 74.0] in the pre-ERAS group (*p* = 0.011). Additionally, the EuroSCORE II was significantly lower in the ERAS group (1.12 [IQR: 0.75; 1.84]) compared to the pre-ERAS group (1.49 [IQR: 0.94; 3.12], *p* < 0.001). The proportion of isolated CABG surgeries was similar between the two groups (30% vs. 30.9%, *p* = 0.884). Other baseline characteristics, including gender, arterial hypertension, diabetes, previous cardiac surgery, and AF (regardless of the type of AF) were comparable between groups. In the matched cohort, differences between the ERAS and pre-ERAS groups were less pronounced. The median age was slightly lower in the ERAS group (63.0 years [IQR: 54.0; 72.0]) than in the pre-ERAS group (66.0 years [IQR: 58.0; 74.0]), but this difference was not statistically significant (*p* = 0.114). The EuroSCORE II values were comparable between groups (1.24 [IQR: 0.73; 2.25] vs. 1.32 [IQR: 0.91; 2.38], *p* = 0.269). A notable difference in surgical type was observed, with isolated CABG being less frequent in the ERAS group (21.4%) than in the pre-ERAS group (32.4%) (*p* = 0.034). Other characteristics, including gender distribution, hypertension, diabetes, prior cardiac surgery, and atrial fibrillation, were similar between groups.

### 3.2. Primary Outcome

In the overall cohort, the incidence of postoperative atrial fibrillation (POAF) was significantly lower in the ERAS group compared to the pre-ERAS group according to both definitions used. Based on the AATS 2014 criteria, POAF occurred in 20% of patients in the ERAS group (76 out of 321) versus 39% in the pre-ERAS group (64 out of 162) (*p* = 0.001). Similarly, using the ESC 2024 criteria, the incidence of POAF was 21% in the ERAS group (57 out of 267) compared to 37% in the pre-ERAS group (51 out of 137) among patients without a prior history of atrial fibrillation (*p* = 0.001). In the matched cohort, these differences remained statistically significant. According to the AATS 2014 criteria, POAF occurred in 23% of patients in the ERAS group (34 out of 145) compared to 39% in the pre-ERAS group (57 out of 145) (*p* = 0.004). Using the ESC 2024 criteria, the incidence of POAF was 20% in the ERAS group (24 out of 119) versus 36% in the pre-ERAS group (45 out of 124) among patients without prior atrial fibrillation (*p* = 0.005) (Table 2).

### 3.3. Preoperative, Intraoperative, and Postoperative Metrics

Prophylaxis for POAF was significantly higher in the ERAS group (66% vs. 22%, *p* = 0.001) than in the overall cohort and remained significantly higher in the matched cohort (70% vs. 21%, *p* = 0.001). Similarly, β-blocker resumption was more frequent in the ERAS group compared to the pre-ERAS group (47% vs. 16%, *p* = 0.001), with a similar pattern observed in the matched cohort (41% vs. 15%, *p* = 0.028). The ERAS group demonstrated significantly shorter CPB durations, aortic cross-clamp times, and extubation times compared to the pre-ERAS group in both overall and matched cohorts (all *p* < 0.001). Mobilization indicators, including mobilization for the first meal and mobilization during POD 1, were notably higher in the ERAS group across all comparisons (*p* < 0.001). Opioid use was consistently reduced in the ERAS group from POD 0 to POD 3, with significant differences (*p* < 0.001). Additionally, bowel recovery was faster in the ERAS group, occurring at a median of 3 days compared to 4.5 days in the pre-ERAS group (*p* < 0.001). Catheters and drains were also removed earlier in the ERAS group. ICU stay durations were relatively short, with a median of approximately 1 day in both groups. Finally, the hospital length of stay (LOS) was significantly reduced, with a median of 9 days compared to 12 days (*p* < 0.001) before the initiation of the program (Table 3).

### 3.4. Complications After Surgery

The incidence of complications, defined as the occurrence of at least one complication among the listed events, was consistently lower in the ERAS group compared to the pre-ERAS group. In the overall cohort, complications occurred in 24% of ERAS patients versus 43% of pre-ERAS patients (*p* < 0.001). This difference remained significant in the matched cohort, with 26% of ERAS patients experiencing complications compared to 38% in the pre-ERAS group (*p* = 0.033). Pleural effusion was particularly reduced in the ERAS group, occurring in 5.61% of patients compared to 17.3% in the pre-ERAS group (*p* < 0.001) for the overall cohort, and 6.9% vs. 15.9% in the matched cohort (*p* = 0.026). Other individual complications displayed lower incidence (reoperation, acute confusional state, hospital-acquired pneumonia) but did not show statistically significant differences between groups (Table 4).

To assess the clinical impact of POAF, we compared postoperative outcomes between patients who developed POAF and those who did not. POAF was significantly associated with higher complication rates, more pleural and pericardial effusions, increased ICU readmissions, and longer hospital stays. No significant difference was observed regarding 30-day mortality (Table 5).

### 3.5. Predictors of POAF: Univariate and Multivariate Analyses

The univariate analysis identified several significant predictors of POAF (according to AATS 2014). Factors associated with an increased risk of POAF included age (OR: 1.05; *p* = 0.001), EuroSCORE II (OR: 1.09; *p* = 0.041), CPB duration (OR: 1.00; *p* = 0.026), and the presence of complications (OR: 2.17; *p* = 0.001). Factors associated with a reduced risk of POAF included isolated CABG surgery (OR: 0.58; *p* = 0.022) and ERAS (OR: 0.47; *p* = 0.001). In the multivariate analysis, which included only variables that were significant in the univariate analysis, the following factors remained independently associated with an increased risk of POAF: age (OR: 1.078; *p* = 0.001), CPB duration (OR: 1.008; *p* = 0.009), and the presence of complications (OR: 1.812; *p* = 0.036). Factors associated with a reduced risk of POAF that remained significant were isolated CABG surgery (OR: 0.786; *p* = 0.026) and ERAS (OR: 0.566; *p* = 0.031) (Table 6). The corresponding forest plot for multivariate analysis is presented in Figure 2.

## 4. Discussion

Our study demonstrates that the implementation of the ERAS protocol in cardiac surgery is associated with a significant reduction in the incidence of POAF according to both AATS 2014 and ESC 2024 definitions. In the overall cohort, POAF occurred in 20% of patients in the ERAS group compared to 39% in the pre-ERAS group (*p* = 0.001). This difference remained significant in the matched cohort, with 23% of ERAS patients experiencing POAF versus 39% in the pre-ERAS group (*p* = 0.004). The same trend was observed using the ESC 2024 criteria among patients without a history of atrial fibrillation, with POAF occurring in 21% of ERAS patients compared to 37% of pre-ERAS patients in the overall cohort (*p* = 0.001), and 20% vs. 36% in the matched cohort (*p* = 0.005). The multivariate analysis identified age, CPB duration, the presence of complications, and ERAS implementation as independent predictors of POAF. Specifically, ERAS was associated with a reduced risk of POAF (OR: 0.566; *p* = 0.031), while older age, longer CPB duration, and the occurrence of complications were associated with increased risk. To further explore the clinical relevance of POAF, we performed an additional analysis comparing outcomes between patients with and without POAF. This analysis confirmed that POAF is significantly associated with an increased risk of postoperative complications, including pleural and pericardial effusions, ICU readmission, delirium, and a prolonged hospital stay. These findings reinforce the negative impact of POAF on recovery after cardiac surgery, consistent with previously published data [9,10,22].

Overall, the implementation of the ERAS program in cardiac surgery was associated with several favorable perioperative outcomes beyond the reduction of POAF. The ERAS group demonstrated shorter CPB durations, aortic cross-clamp times, and extubation times compared to the pre-ERAS group in both the overall and matched cohorts. Additionally, early mobilization was significantly improved, with higher rates of mobilization for the first meal after extubation and during the postoperative period in the ERAS group. Opioid consumption was consistently lower from POD 0 to POD 3 in the ERAS group, indicating better pain management and enhanced recovery. Furthermore, bowel function recovery was faster, and catheters and drains were removed earlier in the ERAS group, contributing to reduced overall postoperative complications and promoting accelerated recovery. Our results regarding reduced CPB and aortic cross-clamp durations align with previous studies reporting significant reductions [30,31,32,33,34].

Additionally, our findings on early extubation are supported by studies reporting an increased likelihood of early extubation with ERAS protocols [35,36]. The enhanced recovery, better postoperative mobility, and bowel function recovery observed in our cohort are also in line with previous findings [34,37]. A reduction in hospital LOS is also consistent with studies that have shown significant improvements [33,34,35,36,38,39,40]. While hospital stays in our center tend to be longer than some international benchmarks (9–12 days vs. 6–7 days), this reflects local discharge protocols, patient characteristics, and healthcare system organization. The 3-day reduction observed with ERAS is a meaningful achievement in our context. Our long-term objective is to further optimize perioperative care and move progressively toward a target average stay of 7 days.

These results corroborate that the ERAS protocol positively influences multiple aspects of postoperative recovery. These overall improvements in perioperative care and the standardized and optimized patient pathway are likely to contribute positively to reducing the incidence of POAF. Enhanced pain control, reduced opioid consumption, early mobilization, and shorter durations of mechanical ventilation, CPB, and aortic cross-clamp times are all known factors that can potentially lower the risk of developing POAF [29,30,31,32,33,41]. Moreover, faster removal of drains and catheters, along with quicker bowel function recovery, can decrease inflammation and reduce physiological stress, further diminishing the risk of atrial fibrillation [34,41]. Therefore, the overall beneficial effects of ERAS protocols on perioperative recovery are expected to have a favorable impact on the incidence of POAF. Many studies have investigated the subject of POAF after cardiac surgery and the impact of the ERAS program. A randomized study by Li et al. included a total of 226 patients and found a positive impact of the ERAS program on the reduction of POAF, with an odds ratio of 0.31 (95% CI: 0.10–1.00) [42]. A prospective observational study by Fleming et al. evaluated the effect of a perioperative care bundle in cardiac surgery involving 105 patients. A significant decrease in the incidence of POAF from 28.3% to 13.5% was observed (*p* = 0.06) [38]. A recent meta-analysis with nine studies and a total of 2131 patients found no statistically significant reduction in the incidence of POAF associated with ERAS programs (OR: 0.77; 95% CI: 0.57–1.03; *p* = 0.08; I^2^ = 17%) [39]. Although there was a remarkable trend towards a reduced incidence of POAF in ERAS groups, the results did not reach statistical significance. The authors identified several limitations of the meta-analysis. Firstly, the majority of the included studies were non-randomized, with only one randomized controlled trial, which limits the robustness of the findings. Secondly, there was considerable heterogeneity in ERAS protocol implementation across studies, complicating the generalizability of the results. Additionally, compliance with ERAS protocols was not consistently reported, and there was substantial variability in how outcomes were defined across studies, which may imply differences in the reported incidence of POAF. In previous studies, the incidence of POAF varies considerably, ranging from 30 to 50% [1,2]. In our cohort, the incidence of POAF ranged from 36% to 39%, depending on the definition applied, which is consistent with the findings of a previous study conducted at our center in 2017 [40]. By contrast, the overall incidence of POAF reported in the studies included in the meta-analysis was relatively low at 23%, which may partly explain the lack of a statistically significant effect of ERAS programs. Regarding differences in POAF definitions across studies that may complicate comparisons with other published data, we utilized two complementary definitions of POAF to provide a comprehensive assessment of postoperative atrial fibrillation. The first definition, derived from the AATS 2014 guidelines, focuses on the practical aspect of POAF by including cases that required specific therapeutic intervention. This approach is relevant in clinical practice as it identifies patients who experienced POAF severe enough to warrant treatment. The second definition, based on the ESC 2024 guidelines, emphasizes a preventive perspective by including all new-onset POAF events, regardless of whether they required intervention. This broader definition allows for a more sensitive detection of POAF and better evaluation of preventive measures. Our study has several limitations that should be acknowledged. The retrospective nature of the 2019 cohort introduces inherent biases, particularly selection bias but also information bias, which may affect the validity of our findings. However, we made every effort to minimize bias by thoroughly collecting all available data from medical records and including all eligible patients who met the inclusion criteria. Additionally, we performed propensity score matching to balance baseline characteristics between the ERAS and pre-ERAS groups, thereby enhancing comparability. The sample size was relatively small, thus potentially weakening the statistical power, although the number of patients included in this study was higher than most studies on the same topic [39]. In the unmatched cohort, both age and EuroSCORE II differed significantly between groups. After performing matching on these variables, the two cohorts were balanced, and the differences became non-significant. Furthermore, multivariate logistic regression adjusting for age and EuroSCORE II confirmed that ERAS implementation remained an independent protective factor against POAF, reinforcing the robustness of our findings. Valvular surgery, especially mitral valve surgery, is known to be more greatly associated with POAF due to atriotomy [8,13,16,35,36]. As shown in Table 1, the demographic characteristics of our study population reveal a substantially higher percentage of valvular surgery in the ERAS cohort (25.9% compared to 41% in the full population, and 27.6% compared to 44.8% in the matched population). This is explained by the fact that we initially start the ERAS program with planned valve surgery [27]. After propensity score matching, the ERAS group included a lower proportion of isolated CABG and a higher proportion of valvular surgeries, which are more strongly associated with POAF. This imbalance may have biased our results conservatively, potentially underestimating the protective effect of the ERAS protocol on POAF incidence. In addition, multivariate regression adjusting for surgical type confirmed that ERAS implementation remained an independent protective factor against POAF, regardless of the procedure performed. Furthermore, the exact classification of preoperative atrial fibrillation (e.g., paroxysmal vs. persistent) was not consistently available, particularly in the retrospective cohort. As a result, a detailed subgroup analysis based on AF subtype could not be performed. As this study focused primarily on the incidence of POAF, detailed data regarding the timing of onset, episode duration, and management strategies were not available in a standardized manner, particularly in the retrospective cohort. This precluded further analysis of AF burden and therapeutic approaches. Although drains were removed earlier in the ERAS group, the incidence of pleural effusion was significantly lower. This may reflect multiple synergistic effects, including shorter CPB times, earlier mobilization, improved pain management, better hemodynamic monitoring, and more effective fluid management strategies. Quantitative drain outputs and routine echocardiographic data were not available for this study, which limits our ability to fully explore this association. Finally, our ERAS protocol did not incorporate certain interventions that have shown potential benefits in reducing POAF. For example, posterior pericardiotomy, a technique that has been reported to significantly reduce the incidence of POAF by facilitating pericardial fluid drainage and reducing inflammation, was not part of our protocol [5,24].

Several pharmacological agents have been shown to affect the risk of POAF occurrence. The most frequently cited agents in the literature are beta-blockers, amiodarone, sotalol, and magnesium [21,22,37,43,44]. Additionally, the literature available concerning corticosteroids, colchicine, nonsteroidal anti-inflammatory drugs (NSAIDs), antioxidant agents, calcium channel blockers, digoxin, angiotensin-converting enzyme inhibitors (ACEIs), and angiotensin II receptor blockers (ARBs) is limited [21,37,44,45]. Data are contradictory regarding statins and glucocorticoids [46]. At present, only beta-blockers and amiodarone are recommended and endorsed by guidelines [21,22]. The choice of beta-blockers in our protocol was based on older guidelines that favored agents such as metoprolol. This long-acting agent is well-suited for preoperative prophylaxis, but we observed that clinicians were concerned about negative hemodynamic impacts in the early postoperative phase, which led to delayed reintroduction. Short-acting agents with lower negative inotropic effects, such as esmolol or especially landiolol, could potentially increase clinicians’ compliance in the postoperative period. Landiolol has demonstrated efficacy in reducing POAF incidence with a favorable safety profile in a meta-analysis of nine Japanese randomized controlled trials with 807 patients and showed a significant reduction in POAF after cardiac surgery (RR = 0.41; 95% CI 0.32–0.52; *p* < 0.001) [47]. Currently, a randomized controlled trial on a European population is ongoing to evaluate the efficacy of landiolol for the prevention of POAF after cardiac surgery (*Landiolol in Postoperative Atrial Fibrillation,* NCT03779178). It is important to note that our protocol was developed and implemented before the release of the most recent ESC 2024 guidelines, which recommend amiodarone as a preferred agent for POAF prophylaxis, particularly in high-risk patients. Future protocols should consider incorporating amiodarone as suggested by the latest guidelines, while acknowledging the need for stronger evidence to support its routine use.

It is important to emphasize that our ERAS protocol is not a definitive or static framework. Following this study, we intend to critically review our protocol. Discussions will be held to consider the introduction of new measures, such as posterior pericardiotomy and intravenous beta-blockers, as well as adapting our choice of prophylactic agents. Continuous assessment of our ERAS protocol will be essential to further improve patient outcomes, optimize perioperative care, and reduce the incidence of POAF.

## 5. Conclusions

In conclusion, implementing a cardiac ERAS program at our institution resulted in a significant reduction in the incidence of POAF, along with improvements in various perioperative parameters that collectively contribute to enhanced patient recovery after cardiac surgery. These findings support the efficacy of the ERAS protocol in optimizing postoperative outcomes. However, further studies, particularly well-designed randomized controlled trials, are needed to confirm these results and strengthen the level of evidence supporting the implementation of ERAS protocols in cardiac surgery.

## Figures and Tables

**Figure 1 biomedicines-13-01212-f001:**
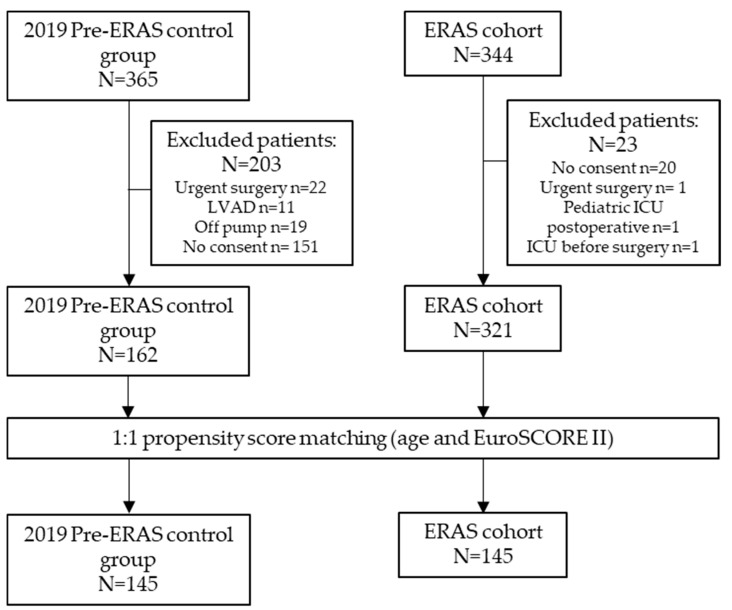
Study flowchart. ERAS: enhanced recovery after surgery; LVAD: left ventricle assist device; ICU: intensive care unit; EuroSCORE II: European System for Cardiac Operative Risk Evaluation II.

**Figure 2 biomedicines-13-01212-f002:**
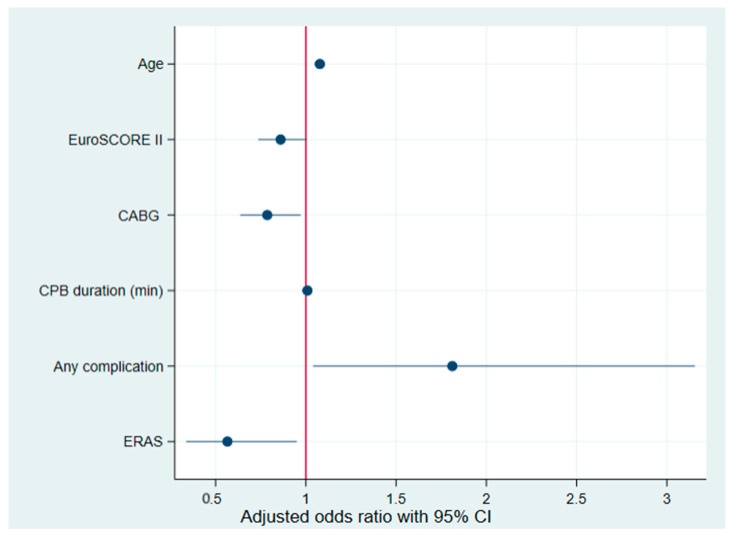
Forest plot of factors influencing the risk of POAF after multivariate analysis. EuroSCORE II: European System for Cardiac Operative Risk Evaluation II; CABG: isolated coronary artery bypass grafting surgery; CPB: cardiopulmonary bypass. ERAS: enhanced recovery after surgery; CI: confidence interval.

**Table 1 biomedicines-13-01212-t001:** Demographic characteristics of the full and matched populations.

Characteristics	All Patients			Matched Patients		
	2019 Pre-ERAS (*n* = 162)	ERAS (*n* = 321)	*p*-Value	2019 Pre-ERAS (*n* = 145)	ERAS (*n* = 145)	*p*-Value
**Age (years)**	67.0 [59.0; 74.0]	64.0 [56.0; 71.0]	0.011	66.0 [58.0; 74.0]	63.0 [54.0; 72.0]	0.114
**Gender (female)**	50 (30.9%)	78 (24.3%)	0.151	44 (30.3%)	36 (24.8%)	0.358
**Previous cardiac surgery**	18 (11.1%)	21 (9%)	0.636	14 (9.66%)	16 (14.2%)	0.355
**LVEF (%)**	60 [55; 65]	60 [55; 65]	0.653	60 [55; 65]	60 [55; 65]	0.964
**Atrial fibrillation**	25 (15%)	54 (16%)	0.458	21 (14%)	26 (17%)	0.529
**Preoperative diabetes**	28 (17.3%)	66 (20%)	0.461	23 (15.9%)	26 (17.9%)	0.754
**Hypertension**	113 (69%)	212 (66%)	0.493	100 (68%)	92 (63%)	0.238
**Dyslipidemia**	78 (48.1%)	192 (59%)	0.019	68 (46.9%)	81 (55.9%)	0.159
**Smoker**	40 (24.7%)	74 (23%)	0.774	36 (24.8%)	30 (20.7%)	0.484
**COPD** **BMI**	25 (15.4%) 25.8.0 [22.2; 30.5]	49 (15.2%) 26.7 [22.7; 29.7]	0.210 0.009	21 (14.4%) 26.1 [23.3; 29.7]	25 (17.2%) 26.3 [23.3; 30.5]	0.813 0.153
**ASA physical status**			0.191			0.050
*Class 2*	14 (8.70%)	37 (11.6%)		14 (9.72%)	12 (8.39%)	
*Class 3*	121 (75.2%)	214 (67.1%)		111 (77.1%)	96 (67.1%)	
*Class 4*	26 (16.1%)	68 (21.3%)		19 (13.2%)	35 (24.5%)	
**EuroSCORE II**	1.49 [0.94; 3.12]	1.12 [0.75; 1.84]	0.001	1.32 [0.91; 2.38]	1.24 [0.73; 2.25]	0.269
**Surgery type**						
*Isolated CABG*	50 (30.9%)	97 (30%)	0.884	47 (32.4%)	31 (21.4%)	0.034
*Valvular surgery*	42 (25.9%)	133 (41%)		40 (27.6%)	65 (44.8%)	
*Aortic root surgery*	22 (13.6%)	38 (11%)		20 (13.8%)	20 (13.8%)	
*CABG + valvular surgery*	21 (13.0%)	24 (7%)		14 (9.66%)	13 (8.97%)	
*Isolated ascending aorta*	12 (7.41%)	15 (4%)		10 (6.90%)	10 (6.90%)	
*Other*	15 (9.26%)	14 (4%)		14 (9.66%)	6 (4.14%)	

ERAS: enhanced recovery after surgery; LVEF: left ventricular ejection fraction; COPD: chronic obstructive pulmonary disease; BMI: body mass index; ASA: American Society of Anesthesiologists; EuroSCORE II: European System for Cardiac Operative Risk Evaluation II; CABG: coronary artery bypass surgery.

**Table 2 biomedicines-13-01212-t002:** Comparison of POAF incidence between ERAS and pre-ERAS groups according to AATS 2014 and ESC 2024 definitions in the overall and matched cohorts.

	All Patients			Matched Patients		
	Pre-ERAS	ERAS	*p*-Value	Pre-ERAS	ERAS	*p*-Value
All patients	*n* = 162	*n* = 321		*n* = 145	*n* = 145	
**POAF AATS 2014**	64 (39%)	76 (20%)	0.001	57 (39%)	34 (23%)	0.004
No AF history	*n* = 137	*n* = 267		*n* = 124	*n* = 119	
**POAF ESC 2024**	51(37%)	57 (21%)	0.001	45 (36%)	24(20%)	0.005

ERAS: enhanced recovery after surgery; POAF: postoperative atrial fibrillation; AATS: American Association for Thoracic Surgery; AF: atrial fibrillation; ESC: European Society of Cardiology.

**Table 3 biomedicines-13-01212-t003:** Comparison of secondary outcomes between the ERAS and pre-ERAS Groups in the overall and matched cohorts.

	All Patients			Matched Patients		
	2019 Pre-ERAS (*n* = 162)	ERAS (*n* = 321)	*p*-Value	2019 Pre-ERAS (*n* = 145)	ERAS (*n* = 145)	*p*-Value
**POAF prophylaxis (%)**	36 (22%)	215 (66%)	0.001	31(21%)	101(70%)	0.001
**β-blocker reinitiation (%)**	27 (16%)	153 (47%)	0.001	23 (15%)	60 (41%)	0.028
**CPB duration (min)**	94.0 [71.0; 125]	72.0 [56.0; 91.0]	<0.001	90.0 [70.8; 124]	77.0 [56.8; 94.8]	<0.001
**Aortic cross-clamp (min)**	71.0 [49.0; 96.8]	53.0 [42.0; 70.0]	<0.001	68.0 [49.0; 95.0]	57.0 [43.0; 73.0]	0.001
**Extubation in OR (%)**	72 (44.4%)	209 (65.3%)	<0.001	69 (47.6%)	92 (63.9%)	0.008
**Extubation time (hours)**	2.15 [0.00; 5.65]	0.00 [0.00; 3.53]	<0.001	1.25 [0.00; 4.50]	0.00 [0.00; 4.03]	0.024
**Opioid use (MME)**						
*POD 0*	20.0 [9.25; 38.0]	14.0 [7.50; 22.6]	<0.001	19.0 [8.00; 36.0]	15.0 [7.50; 23.0]	0.022
*POD 1*	23.9 [14.9; 46.8]	16.5 [10.6; 27.8]	<0.001	22.3 [13.9; 44.8]	19.0 [10.8; 29.7]	0.025
*POD 2*	12.4 [5.00; 19.8]	3.30 [0.00; 10.0]	<0.001	12.4 [5.00; 19.8]	3.30 [0.00; 9.90]	<0.001
*POD 3*	5.00 [0.00; 12.4]	0.00 [0.00; 0.00]	<0.001	5.00 [0.00; 12.4]	0.00 [0.00; 0.00]	<0.001
**Mobilization for first meal**	4 (2.55%)	142 (44.4%)	<0.001	4 (2.86%)	71 (49.0%)	<0.001
**Mobilization POD 1 (%)**	77 (59.2%)	224 (83.9%)	<0.001	76 (63.9%)	98 (81.7%)	0.003
**Drainage duration (days)**	3.85 ± 0.40	2.41 ± 0.44	0.039	3.73 ± 0.43	2.79 ± 0.25	0.060
**CVC duration (days)**	5.39 ± 0.45	3.86 ± 0.46	0.041	5.43 ± 0.50	4.34 ± 0.28	0.061
**Bowel recovery (days)**	4.50 [4.00; 5.00]	3.00 [3.00; 4.00]	<0.001	4.50 [4.00; 5.00]	3.00 [3.00; 4.00]	<0.001
**ICU stay (days)**	1.08 [0.94; 2.11]	1.11 [0.95; 1.99]	0.626	1.08 [0.94; 2.02]	1.10 [0.94; 1.92]	0.681
**LOS (days)**	12.0 [9.00; 16.0]	9.00 [7.00; 13.0]	<0.001	12.0 [8.00; 15.0]	9.00 [8.00; 12.0]	<0.001

POAF: postoperative atrial fibrillation; CPB: cardiopulmonary bypass; OR: operating room; MME: morphine milligram equivalent; POD: postoperative day; CVC: central venous catheter; β-blocker: beta-blocker; ICU: intensive care unit; LOS: length of hospital stay.

**Table 4 biomedicines-13-01212-t004:** Comparison of postoperative complications between the ERAS and pre-ERAS Groups in the overall and matched cohorts.

Variables	All Patients			Matched Patients		
	2019 Pre-ERAS (*n* = 162)	ERAS (*n* = 321)	*p*-Value	2019 Pre-ERAS (*n* = 145)	ERAS (*n* = 145)	*p*-Value
**Any complications**	70 (43%)	78 (24%)	<0.001	56 (38%)	39 (26%)	0.033
**30-day mortality**	3 (1.85%)	1 (0.31%)	0.112	1 (0.69%)	1 (0.69%)	1.000
**Reoperation (%)**	15 (9.26%)	22 (6.94%)	0.472	12 (8.28%)	13 (9.09%)	0.971
**ICU readmission (%)**	7 (4.32%)	13 (4.10%)	1.000	5 (3.45%)	8 (5.59%)	0.553
**Acute confusional state (%)**	13 (8.02%)	15 (4.67%)	0.200	11 (7.59%)	6 (4.14%)	0.317
**Stroke (%)**	4 (2.47%)	14 (4.36%)	0.434	3 (2.07%)	7 (4.83%)	0.334
**Postoperative MI (%)**	1 (0.62%)	1 (0.31%)	1.000	1 (0.69%)	0 (0.00%)	1.000
**HAP (%)**	21 (13.0%)	26 (8.10%)	0.124	17 (11.7%)	8 (5.52%)	0.094
**Other hospital-acquired infections (%)**	12 (7.41%)	15 (4.67%)	0.305	10 (6.90%)	7 (4.83%)	0.617
**Pleural effusion (%)**	28 (17.3%)	18 (5.61%)	<0.001	23 (15.9%)	10 (6.90%)	0.026
**AKI requiring dialysis (%)**	2 (1.23%)	6 (1.87%)	0.724	2 (1.38%)	4 (2.76%)	0.684

ERAS: enhanced recovery after surgery; ICU: intensive care unit; MI: myocardial infarction; HAP: hospital-acquired pneumonia; AKI: acute kidney injury.

**Table 5 biomedicines-13-01212-t005:** Comparison of postoperative outcomes between patients with POAF and patients without POAF.

Outcome	No POAF (*n* = 341)	POAF (*n* = 142)	*p*-Value
**ICU readmission**	10 (2.9%)	10 (7.2%)	0.035
**Reoperation**	20 (5.9%)	17 (11.9%)	0.020
**Pleural effusion**	26 (7.5%)	20 (14.2%)	0.023
**Pericardial effusion**	6 (1.7%)	10 (7.1%)	0.003
**Acute confusional state**	10 (2.9%)	18 (12.8%)	0.000
**Hospital stay**	9 [7–12.8]	12 [9–18]	0.034
**30-day mortality**	3 (0.8%)	1 (0.7%)	0.860

POAF: postoperative atrial fibrillation; ICU: intensive care unit.

**Table 6 biomedicines-13-01212-t006:** Univariate and multivariate logistic regression analyses for predictors of POAF.

Variables	Univariate OR (95% CI)	*p*-Value	Multivariate OR (95% CI)	*p*-Value
**Age**	1.05 (1.03–1.07)	0.001	1.078 (1.0489–1.1070)	0.001
**Gender**	1.10 (0.70–1.71)	0.666	-	-
**BMI**	1.02 (0.98–1.06)	0.220	-	-
**Diabetes**	0.95 (0.76–1.19)	0.703	-	-
**Hypertension**	0.98 (0.92–1.05)	0.697	-	-
**COPD**	0.95 (0.85–1.07)	0.482	-	-
**EuroSCORE II**	1.09 (1.00–1.20)	0.041	0.860 (0.7365–1.0047)	0.057
**LVEF**	0.99 (0.97–1.01)	0.419	-	-
**Isolated CABG**	0.58 (0.37–0.92)	0.022	0.786 (0.6361–0.9714)	0.026
**CPB Duration (min)**	1.00 (1.00–1.00)	0.026	1.008 (1.0021–1.0149)	0.009
**Any Complication**	2.17 (1.43–3.28)	0.001	1.812 (1.0402–3.1549)	0.036
**ERAS**	0.47 (0.31–0.71)	0.001	0.566 (0.3369–0.9493)	0.031

OR: odds ratio; CI: confidence interval; BMI: body mass index; COPD: chronic obstructive pulmonary disease; EuroSCORE II: European System for Cardiac Operative Risk Evaluation II; LVEF: left ventricular ejection fraction; CABG: coronary artery bypass grafting; CPB: cardiopulmonary bypass; ERAS: enhanced recovery after surgery.

## Data Availability

Data can be obtained upon request to the corresponding author.

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
