# Peer review of "Effect of Enhanced Recovery After Surgery (ERAS) Implementation on Postoperative Atrial Fibrillation in Cardiac Surgery"

_biomedicines, 2025, doi:10.3390/biomedicines13051212_

Round 1

Reviewer 1 Report

Comments and Suggestions for Authors

In this monocentric, observational cohort study, the authors addressed an important topic and compared a retrospective pre-ERAS cohort (n=162) with a prospective ERAS cohort 20 (n=321). The primary outcome was the incidence of POAF, assessed using two definitions: 21 (1) the American Association for Thoracic Surgery (AATS) 2014 clinical definition, identifying POAF as atrial fibrillation requiring treatment; and (2) the European Society of Cardiology (ESC) 2024 definition, describing new-onset atrial fibrillation occurring immediately after surgery. Secondary outcomes included compliance with POAF prophylaxis measures, length of hospital stay, and the occurrence of postoperative complications. Statistical analyses included propensity score matching and multivariate logistic regression to identify independent predictors of POAF. 

Major findings. The study is well conducted and the results of clinical interest. Some points should be underscored or added at the limitations section.

  • Major limitation
  1. Isolated CABG n=50 (30.9%) n=97 (30%) p= 0.884; n= 47 (32.4%) n= 31 (21.4%); p=0.034 ; Valvular surgery n= 42 (25.9%) n= 133 (41%) n=40 (27.6%) n=65 (44.8%); Valvular surgery is known to be greater associated with post op AFib; p=0.034;  this point is a major point of discussion as well as  the lower % of isolated CABG group when propensity matched score was performed ???
  2. The % of paroxysmal or persistent AFib in each population should be available regarding the prognostic impact of persistent AFib requiring many other treatments;
  3. Many another drugs may affect the risk of post op AFib occurence; this element should be improved 
  4. The introduction appears too long

Reviewer 2 Report

Comments and Suggestions for Authors

I have carefully read the article titled "Effect of Enhanced Recovery After Surgery (ERAS) Implementation on Postoperative Atrial Fibrillation in Cardiac Surgery" sent to me for evaluation. My comments, criticisms and suggestions are listed below:
1. Enhanced Recovery After Surgery (ERAS) programs have been shown to reduce surgical injury, support recovery and improve postoperative clinical outcomes in patients undergoing cardiac surgery.
2. Well-designed studies are needed regarding this application, which has become increasingly popular in recent years. Therefore, the current study is valuable.
3. Although the number of cases is limited, it was possible to obtain statistically significant data since it evaluated two groups of patients with similar characteristics.
4. It would be appropriate to provide details about the onset, duration and treatments applied to postoperative atrial fibrillation in both groups of patients in the study and to discuss the effects of atrial fibrillation on clinical outcomes.
4. According to the authors' own statements, the patients in the ERAS group were younger and had lower Euroscore-II values. In the univariate and multivariate analyses in the article, age was determined to be one of the most determining factors in the development of postoperative atrial fibrillation. It would be appropriate to comment on the effect of this situation on the results.

5. Is there a difference in terms of surgical approach between the two groups? (such as minimal or less invasive, robotic surgery). If so, did this affect the results?

6. Is there a difference in body surface area or body mass index between the two groups of patients?

7. Are there any precautions taken in patients with anemia in the preoperative period? What is the approach to postoperative anemia? Is there a difference between the groups in terms of these parameters?

8. The amount of postoperative bleeding is not included in the article. According to the authors' statements, catheters and drains were removed earlier in the ERAS group. The presence of postoperative pericardial effusion is a predisposing factor for postoperative atrial fibrillation. Therefore, discussing the drainage amounts of the two groups and the postoperative echocardiography findings, if any, will be explanatory. The reason why the amount of pleural effusion was less in the ERAS group despite the earlier removal of the drains should be discussed.

9. Many cardiac surgery patients can be discharged in an average of 6-7 days. According to the authors, the average discharge time for patients in both groups is 9-12 days. Is the reason for these periods related to the protocols of the center where the authors work or to surgeon preferences?

Best regards

Round 2

Reviewer 1 Report

Comments and Suggestions for Authors

No further comments. The authors have replied where required.